SOFTWARE

# BCI Toolbox: An open-source python package for the Bayesian causal inference model

**Haocheng Zhu[1,2], Ulrik Beierholm[3], Ladan Shams****[1,4]\***

**1** Department of Psychology, University of California, Los Angeles, California, United States of America, **2** Department of Psychology, Research Center for Psychology and Behavioral Sciences, Soochow University, Suzhou, China, **3** Department of Psychology, University of Durham, Durham, United Kingdom, **4** Department of Bioengineering, and Neuroscience Interdepartmental Program, University of California, Los Angeles, California, United States of America

\* ladan@psych.ucla.edu

## Abstract

Psychological and neuroscientific research over the past two decades has shown that the Bayesian causal inference (BCI) is a potential unifying theory that can account for a wide range of perceptual and sensorimotor processes in humans. Therefore, we introduce the BCI Toolbox, a statistical and analytical tool in Python, enabling researchers to conveniently perform quantitative modeling and analysis of behavioral data. Additionally, we describe the algorithm of the BCI model and test its stability and reliability via parameter recovery. The present BCI toolbox offers a robust platform for BCI model implementation as well as a hands-on tool for learning and understanding the model, facilitating its widespread use and enabling researchers to delve into the data to uncover underlying cognitive mechanisms.

**Data Availability Statement:** We shared our code and test dataset on https://github.com/evans1112/bcitoolbox.

## Introduction

It has been proposed that the human brain functions like a Bayesian statistical machine [1], with the nervous system continuously processing uncertain sensory information from different modalities to infer the causes of sensory observation. The Bayesian Causal Inference (BCI) model [2] is a normative Bayesian framework that describes this process, wherein inferences are made regarding both the causal structure (common cause vs. independent causes) and the sources of the sensory inputs. In the BCI model, these inferences are coherently unified, involving a competition between two hypotheses–were the sensory information generated by a common cause or by independent causes–to account for observed sensory measurements.

During the past two decades, the BCI model has been extended and employed in a large variety of perceptual and sensorimotor domains [3,4], including temporal numerosity judgment [5,6], spatial localization judgment [2,7–9], size-weight illusion paradigm [10], rubber-hand illusion paradigm [11–13], and heading perception [14]. Given this empirical evidence, the BCI model has been recognized as a potential unifying framework in neuroscience [4]. Meanwhile, computational modeling methods have provided a new perspective to psychiatric research as well [15–17], demonstrating how these models can deepen our understanding of the pathophysiological processes underlying mental disorders and inform therapeutic

**Funding:** The author(s) received no specific funding for this work.

**Competing interests:** The authors have declared that no competing interests exist.

interventions. Noel and colleagues emphasized the importance of causal inference in computational psychiatry [18,19].

Inspired by the substantial potential of the BCI model and the increasing demand for Bayesian data analysis within the domain of neuroscience and psychology, we introduce the Bayesian causal inference toolbox (BCI Toolbox), a zero-programming software package written in Python, to the scientific community as a tool for understanding and using the BCI model. The BCI Toolbox features a graphical user interface (GUI) for primary use and well-studied mathematical functions for advanced use. To facilitate the use of the BCI model, the GUI includes user-friendly model fitting and simulation functionalities. The software can be installed from the online documentation (https://bcitoolboxrmd.readthedocs.io/en/latest/index.html), GitHub (https://github.com/evans1112/bcitoolbox), or via PIP (https://pypi.org/project/bcitoolbox).

Here, we provide an overview of the algorithm implementation and software architecture of the BCI Toolbox and discuss the performance of the BCI model through parameter recovery, further corroborating the model's reliability.

## Design and implementation

In principle, the general implementation is based on the Bayesian causal inference model of multisensory perception [2]. To describe the basic structure of the model, we use the example of hearing a sound and seeing a sight while estimating the location of the sound ($s_A$). However, the model is general and not specific to any sensory modalities or perceptual tasks. The toolbox implementation allows for the combination of two sensations from any modalities and supports a variety of perceptual tasks, as discussed in the following sections.

Fig 1A shows the generative model of BCI, wherein two possible causal structures, namely a common cause and independent causes, can give rise to sensory inputs $x_A$ and $x_V$. During the inference stage of perception, these two hypotheses compete to explain the sensory observations in order to estimate the perceptual variables of interest, e.g., the location of the auditory event ($s_A$) and the location of the visual event ($s_V$). As shown in Fig 1B, the underlying causal structure of the stimuli is inferred based on the available sensory evidence and prior knowledge. Each stimulus or event $s$ in the world causes a noisy sensation $x_i$ of the event. We use the generative model to simulate experimental trials and subject responses by performing Monte Carlo simulations. Each sensation is modeled using the likelihood function $p(x_i|s)$. Trial-to-trial variability is introduced by sampling from a normal distribution around the true locations $s_A$ and $s_V$, plus bias terms $\gamma_A$ and $\gamma_V$ for auditory and visual modalities, respectively [9]. This simulates the corruption of auditory and visual sensory channels by independent Gaussian noise with standard deviation $\sigma_A$ and $\sigma_V$, respectively. In other words, the sensations $x_A$ and $x_V$ are simulated by sampling from the distributions shown in Eqs 1 and 2.

$$x_A \sim N(s_A + \gamma_A, \sigma_A) \tag{1}$$

$$x_V \sim N(s_V + \gamma_V, \sigma_V) \tag{2}$$

We assume there is a prior bias for the sensory information [20], modeled by a Gaussian distribution centered at $\mu_P$. The standard deviation of the Gaussian, $\sigma_P$, determines the strength of the bias. Therefore, the prior distribution of sensory information is:

$$p(s) = N(\mu_P, \ \sigma_P) \tag{3}$$

As the causal structure is unknown to the nervous system, it must be inferred using sensory information and prior knowledge. The probability of each causal structure is computed using

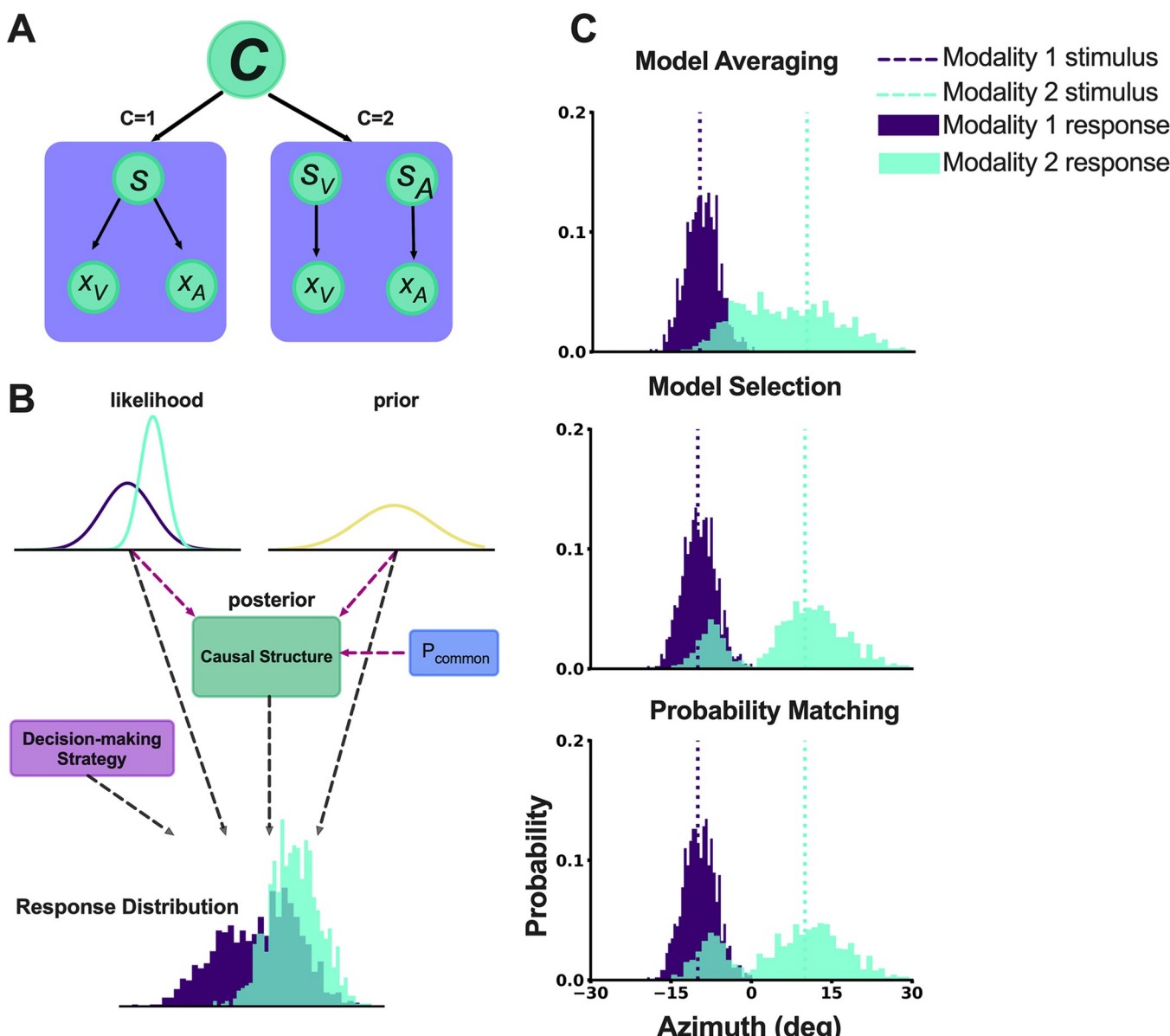

**Fig 1. The general structure of the BCI model and simulation results of BCI Toolbox.** (A) The generative model of BCI, assumes that there is either one cause (C = 1) or two causes (C = 2), leading to the creation of the perceptual variables (s or $s_A$ and $s_V$). (B) The structure of the hierarchical BCI model in the BCI Toolbox. The causal structure is inferred by combining sensory likelihood and prior (prior stimulus expectation and $p_{common}$. $p_{common}$ represents *a priori* expectation of a common cause). The observer response is based on the inferred causal structure, and the decision-making strategy. (C) The one-dimensional model simulation results (generated by $p_{common}$ = 0.5; $\sigma_1$ = 3; $\sigma_2$ = 8; $\sigma_P$ = 30; $\mu_P$ = 0; $s_1$ = -10; $s_2$ = 10) from 3 different decision-making strategy using the BCI Toolbox.

Bayes Rule as follows:

$$p(C|x_A, x_V) = \frac{p(x_A, x_V|C)p(C)}{p(x_A, x_V)}$$ (4)

The optimal estimate of source *s* in each modality depends on the causal structure. If the sensations are produced by independent causes, the estimate of *s* is a weighted average of the

unisensory signal and the prior for $s$:

$$\hat{s}_{(A,C=2)} = \frac{\frac{x_A}{\sigma_A^2} + \frac{x_P}{\sigma_P^2}}{\frac{1}{\sigma_A^2} + \frac{1}{\sigma_P^2}}, \ \hat{s}_{(V,C=2)} = \frac{\frac{x_V}{\sigma_V^2} + \frac{x_P}{\sigma_P^2}}{\frac{1}{\sigma_V^2} + \frac{1}{\sigma_P^2}} \tag{5}$$

If the sensations are produced by a common cause, the estimate of $s$ is a weighted average of the both sensory signals and the prior for $s$:

$$\hat{s}_{(A,C=1)} = \hat{s}_{(V,C=1)} = \frac{\frac{x_A}{\sigma_A^2} + \frac{x_V}{\sigma_V^2} + \frac{x_P}{\sigma_P^2}}{\frac{1}{\sigma_A^2} + \frac{1}{\sigma_V^2} + \frac{1}{\sigma_P^2}} \tag{6}$$

As can be seen in Eq (4), the inference about the causal structure is probabilistic, and therefore, there is uncertainty associated with each causal structure. The optimal estimate of the sources $s_A$ and $s_V$ depend on the goal of the perceptual system in a given task. If the goal is to minimize the average error in the magnitude of the source estimates, i.e., a sum squared error cost function, then optimal strategy for achieving this goal is *model averaging*, in which optimal estimates corresponding to both causal structures are taken into account, however, proportional to their respective probability [2,7].

$$\hat{s}_A = p(C = 1|x_A, x_V)\hat{s}_{(A,C=1)} + p(C = 2|x_A, x_V)\hat{s}_{(A,C=2)}$$

$$\hat{s}_V = p(C = 1|x_A, x_V)\hat{s}_{(V,C=1)} + p(C = 2|x_A, x_V)\hat{s}_{(V,C=2)} \tag{7}$$

However, there are other plausible cost functions. Indeed, Wozny et al. [7] showed that in a spatial localization task many observers' performance was more consistent with *model selection* or *probability matching* strategies. If the nervous system's goal is to optimize the inference of causal structure, this would result in a decision strategy that selects the causal structure with the highest posterior probability and estimates the sensory sources entirely based on the selected causal structure. (Eq 8).

$$\hat{s}_A = \begin{cases} \hat{s}_{(A,C=1)} & \text{if } p(C = 1|x_A, x_V) > 0.5 \\ \hat{s}_{(A,C=2)} & \text{if } p(C = 1|x_A, x_V) \leq 0.5 \end{cases}$$

$$\hat{s}_V = \begin{cases} \hat{s}_{(V,C=1)} & \text{if } p(C = 1|x_A, x_V) > 0.5 \\ \hat{s}_{(V,C=2)} & \text{if } p(C = 1|x_A, x_V) \leq 0.5 \end{cases} \tag{8}$$

*Probability matching* is a stochastic decision-making strategy wherein the nervous system computes the posterior probabilities of potential causal structures (Eq 9). Subsequently, a probabilistic selection mechanism is employed, whereby a decision regarding the endorsement of either a common-cause or independent-cause hypothesis is made stochastically, as shown in Eq 9.

This strategy is optimal if learning is also a factor in the utility function [7].

$$\hat{s}_A = \begin{cases} \hat{s}_{(A,C=1)} \ if \ p(C=1|x_A,x_V) > \xi \\ \hat{s}_{(A,C=2)} \ if \ p(C=1|x_A,x_V) \le \xi \end{cases}$$

$$\hat{s}_V = \begin{cases} \hat{s}_{(V,C=1)} \ if \ p(C=1|x_A,x_V) > \xi \\ \hat{s}_{(V,C=2)} \ if \ p(C=1|x_A,x_V) \le \xi \end{cases} \tag{9}$$

$\xi \in [0:1]$ uniform distribution and sampled on each trial.

More details on the model can be found elsewhere [2,8]; we point the interested reader to earlier publications for additional information.

## Graphical User Interface (GUI)

To enhance user experience, we created an intuitive Graphical User Interface (GUI) specifically for researchers. Fig 2 depicts the overall layout of the interface. The GUI currently provides two core functions: model simulation and model fitting, both of which are detailed below. The GUI supports fitting partial data, for example, behavioral data where only single-modality information is reported. However, it is important to note that the reliability of these fitted parameters may be diminished in cases of partial data.

## Model fitting

In this module, users can input behavioral data for model fitting. The BCI Toolbox supports fitting two types of data: discrete and continuous. It can either maximize the likelihood of the data given the model (equivalent to minimizing the negative likelihood) or minimize the squared error between the model and data. Users have the flexibility to choose various decision strategies, each associated with a different cost function [7]. Additionally, they can adjust seven key parameters to customize the BCI model to their experimental paradigm. These parameters include $p_{common}$ (the prior expectation of a common cause), $\mu_1$, $\mu_2$ (the mean of the likelihood), $\sigma_1$, $\sigma_2$ (the standard deviation of the likelihood), $\gamma_1$ and $\gamma_2$ (perceptual bias). Each parameter can be specified as a free parameter, or can be manually fixed at a value. The GUI's built-in plotting functions enable users to visualize the fitting results.

The user can choose between two methods for parameter optimization: a) the 'Powell' algorithm from the 'minimize'function in the *scipy* package (https://scipy.org) or b) the 'VBMC'method from the *pyvbmc* package (https://acerbilab.github.io/pyvbmc/). The latter is an approximate Bayesian inference method designed for fitting computational models with a limited budget of potentially noisy likelihood evaluations. This makes it particularly useful for computationally expensive models or for quick inference and model evaluation [21–23].

## Model simulation

The simulation function enables users to explore the effects of different parameter values on behavioral outcomes (i.e., response distributions), helping them develop an intuitive understanding of BCI. It also facilitates the investigation of BCI behavior under various parameter or stimulus conditions, which is valuable for qualitatively comparing empirical data with the model. In cases where data sets are missing or limited and reliable fitting isn't possible, model simulation can be used to qualitatively compare empirical and simulated data patterns. The BCI Toolbox provides five parameters for use: $p_{common}$, $\sigma_1^2$ (controlling the variance of modality 1 likelihood), $\sigma_2^2$ (controlling the variance of modality 2 likelihood) $\sigma_p^2$ (controlling the variance

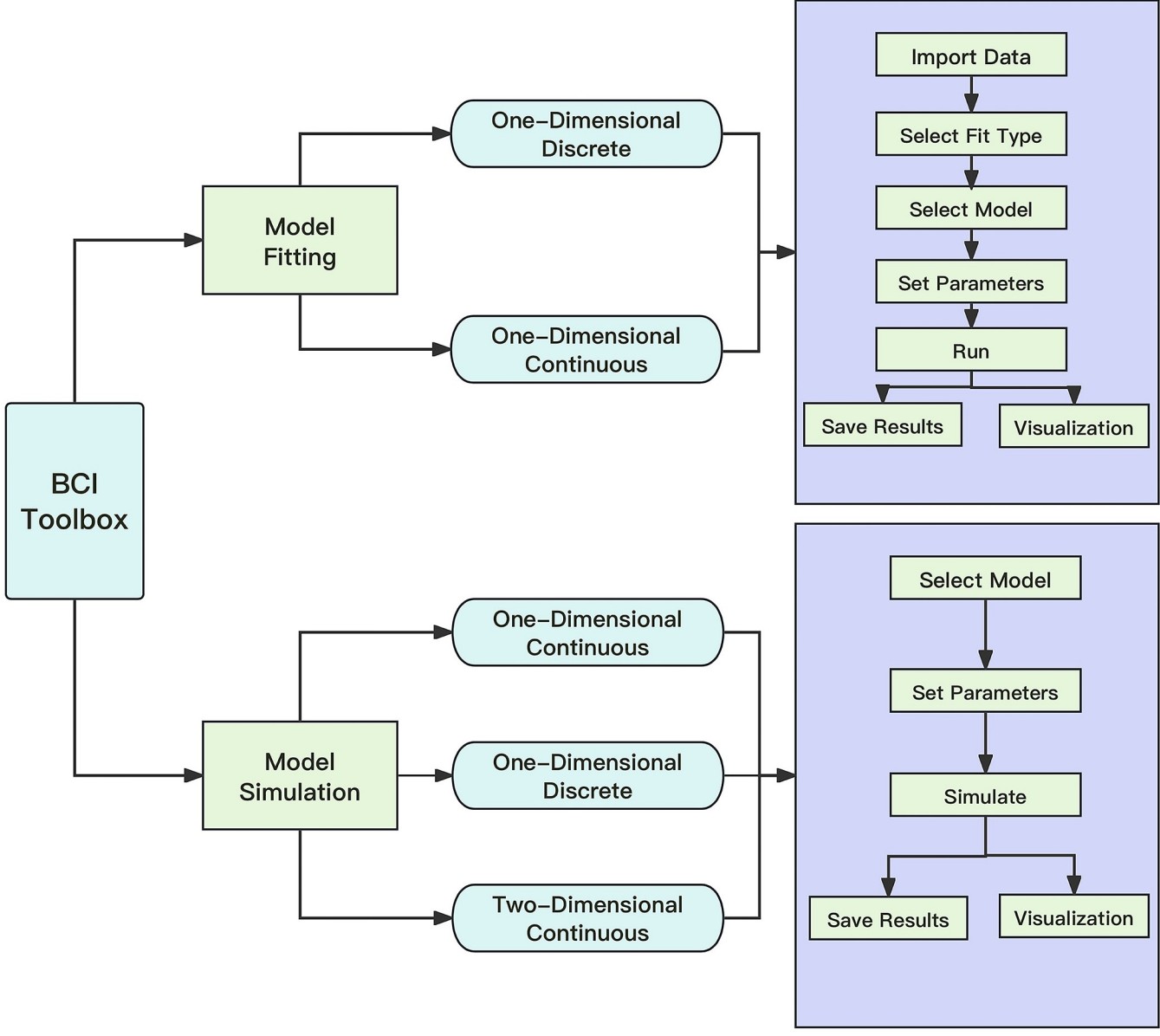

**Fig 2. An overview of the BCI Toolbox GUI.** The GUI provides two main functions: model fitting and model simulation. In the model fitting section, the GUI incorporates two data types: discrete and continuous data. In the model simulation section, the GUI incorporates one-dimensional and two-dimensional simulations. For more details, see BCI Toolbox documentation: https://bcitoolboxrmd.readthedocs.io/.

of prior for perceptual variable of interest) and $\mu_P$ (the mean of the prior for perceptual variable of interest). Upon setting these parameters and choosing the value of stimuli (e.g., the location of each stimulus in a localization task) to observe, the toolbox generates and visualizes the simulated data. Researchers can examine the resulting data and compare the perceptual responses under the three decision-making strategies (*model averaging* vs. *model selection* vs. *probability matching*, Fig 1C). The model simulation module supports one-dimensional continuous data simulation, two-dimensional continuous data simulation and one-dimensional discrete data simulation.

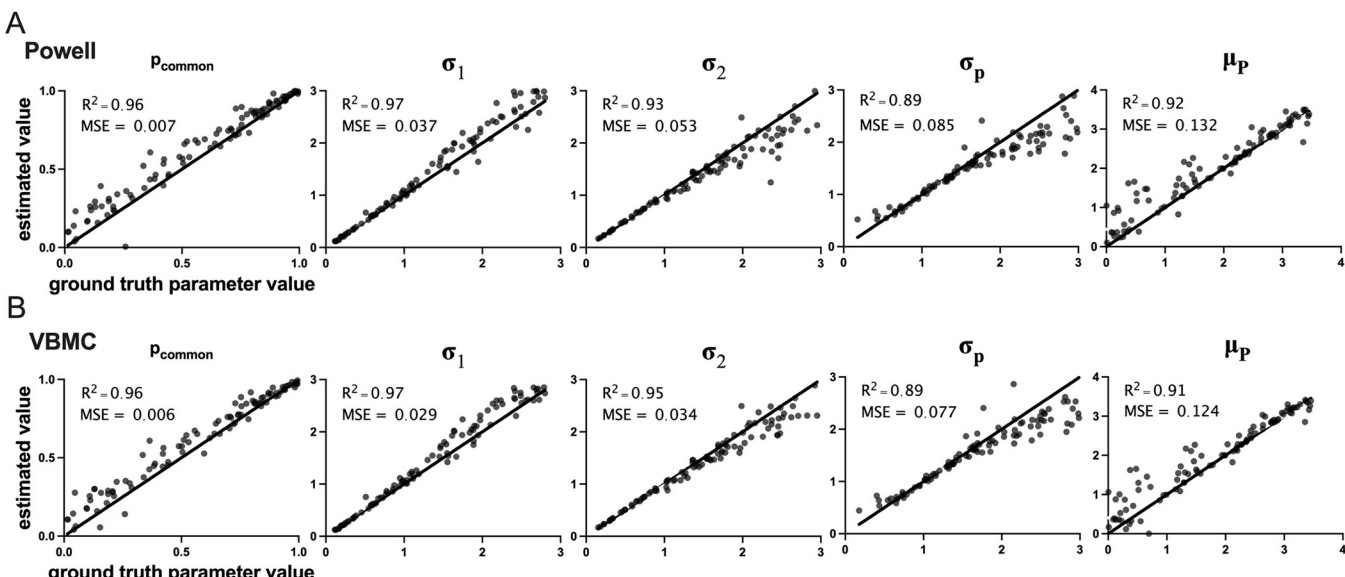

**Fig 3. Results of parameter recovery analysis.** We generated 100 sets of synthetic data under 15 conditions by selecting random values for the 5 model parameters using the discrete 1-dimensional model simulation module of the toolbox. Next, the synthetic data were fitted by the data fitting module of the toolbox. In each panel, the estimated parameter value from data fitting is plotted against the ground-truth value of that parameter. $R^2$ indicates the degree of correlation between the estimated and true parameters. MSE indicates the mean of squared error between data and identity lines (solid lines). In all cases, the model parameters were recovered well. (A) Results from using the Powell algorithm for parameter optimization. (B) Results from using the VBMC method for parameter optimization.

### Parameter recovery

The reliability of the BCI model was assessed through parameter recovery tests. To evaluate the performance of our model, we first simulate synthetic data with known ground truth parameters. We generated a set of 5 random parameters, where $p_{common} \sim U(0,1)$, $\sigma_1 \sim U(0.1,3)$; $\sigma_2 \sim U(0.1,3)$; $\sigma_P \sim U(0.1,3)$; $\mu_P \sim U(0,3.5)$. Subsequently, we generated synthetic data using these parameters under the discrete data model structure (for numerosity task, non-negative) and through *model averaging*.

Next, we used the BCI toolbox data fitting module to fit the model to the data. Using 10,000 as the *number of simulations, minus log likelihood* as the fit type, and *model averaging* as a strategy, we fitted the synthetic data once using the Powell algorithm and once using the VBMC method. Fig 3 shows the results of parameter optimization for each of these two methods.

### Results

Here, for illustration purposes, we present an example of the results produced by the toolbox for each of its two main functions: data fitting and model simulation. For the data fitting, we used data from Experiment 5 in the study by Odegaard et al. [24], which is publicly available. In this experiment, observers were presented with simple visual and auditory stimuli at one of 5 possible positions along the azimuth and were asked to report the perceived location of each stimulus in each trial. The responses were provided using a joystick along a continuous horizontal scale on the screen, making them continuous data. In that experiment, participants' spatial localization was tested using a test session as described above. Following the test session, participants were passively presented with auditory-visual stimuli in an "adaptation" phase. Immediately after the adaptation phase, the participants were tested again in a spatial localization test identical to the pre-adaptation test. The study reported a statistically-

significant increase in $p_{common}$ after adaptation. We analyzed the behavioral data from the spatial localization tasks using the continuous one-dimensional data fitting module with 5 free parameters ($p_{common}$, $\sigma_1$, $\sigma_2$, $\sigma_P$ and $\mu_P$), the Powell parameter optimization method, and *model averaging*. Fig 4A shows the results of data fitting for the pre-adaptation test. The toolbox results replicated the finding of the Odegaard et al. [24] study, including the increase in $p_{common}$ after adaptation (S1 Text).

Fig 4B shows the results of model simulation under four different parameter regimes in the one-dimensional discrete module. As an example, we considered the temporal-numerosity task, where the observer's task is to report the perceived number of flashes and beeps. The responses in this task are discrete. The number of flashes was set to 1 and the number of beeps was set to 2. In the first row, we show the effect of $p_{common}$ on the responses by keeping all other parameters the same but changing the value of $p_{common}$. The left panel in the top row shows the results for a small $p_{common}$ value (low tendency to integrate), whereas the right panel in the top row shows the results for a large $p_{common}$ value (high tendency to integrate). Given the higher precision of the auditory modality, the visual perception (perceived number of flashes) is biased by the number of beeps. However, the degree of bias is influenced by the value of $p_{common}$, with a stronger bias (more illusion) observed in the case of higher tendency for integration. The bottom row shows the effect of visual precision ($\sigma_V$) on the responses, by keeping all parameters the same but changing the value of $\sigma_V$. Lower visual precision leads to a stronger bias and a more pronounced illusion in the visual modality. These example simulations illustrate BCI Toolbox's utility as a tool for understanding the model and predicting behavioral outcomes in different settings

## Availability and future directions

We introduced a toolbox for the Bayesian causal inference model that supports the analysis and simulation of behavioral data across a wide range of tasks in multisensory perception and sensorimotor science. BCI Toolbox provides modeling tools for diverse experimental paradigms and data types, offering various computational and optimization methods within the Bayesian framework. Additionally, it can batch-process and visualize analysis results, enhancing the understanding and practical application of the BCI model.

### Major advantages of BCI Toolbox

One of BCI Toolbox's primary advantages is its user-friendly GUI, which enables and facilitates the use of hierarchical Bayesian causal inference models in neuroscience research, even for researchers without computational training. The BCI Toolbox is suitable for new users to learn and utilize the BCI model. The simulation section can be used for pedagogical purposes, allowing users to intuitively understand the role of various parameters, which can help them further understand the algorithm in the model. Moreover, the simulation functions are useful for qualitative modeling, offering insights into system behaviors beyond mere reliance on quantitative data.

Besides model simulation, the model fitting module is helpful in the quantitative analysis of data, enabling precise parameter estimation and ensuring a more accurate representation of underlying trends behind the behavioral data. The toolbox also offers a model comparison option (using Bayesian Information Criterion, BIC) that enables a thorough comparison of the three decision strategies. Additionally, it supports evaluations based on varying numbers of free parameters.

In addition to the functional advances, we verified the reliability of the BCI model through parameter recovery. The results show that the vast majority of the parameters can be well

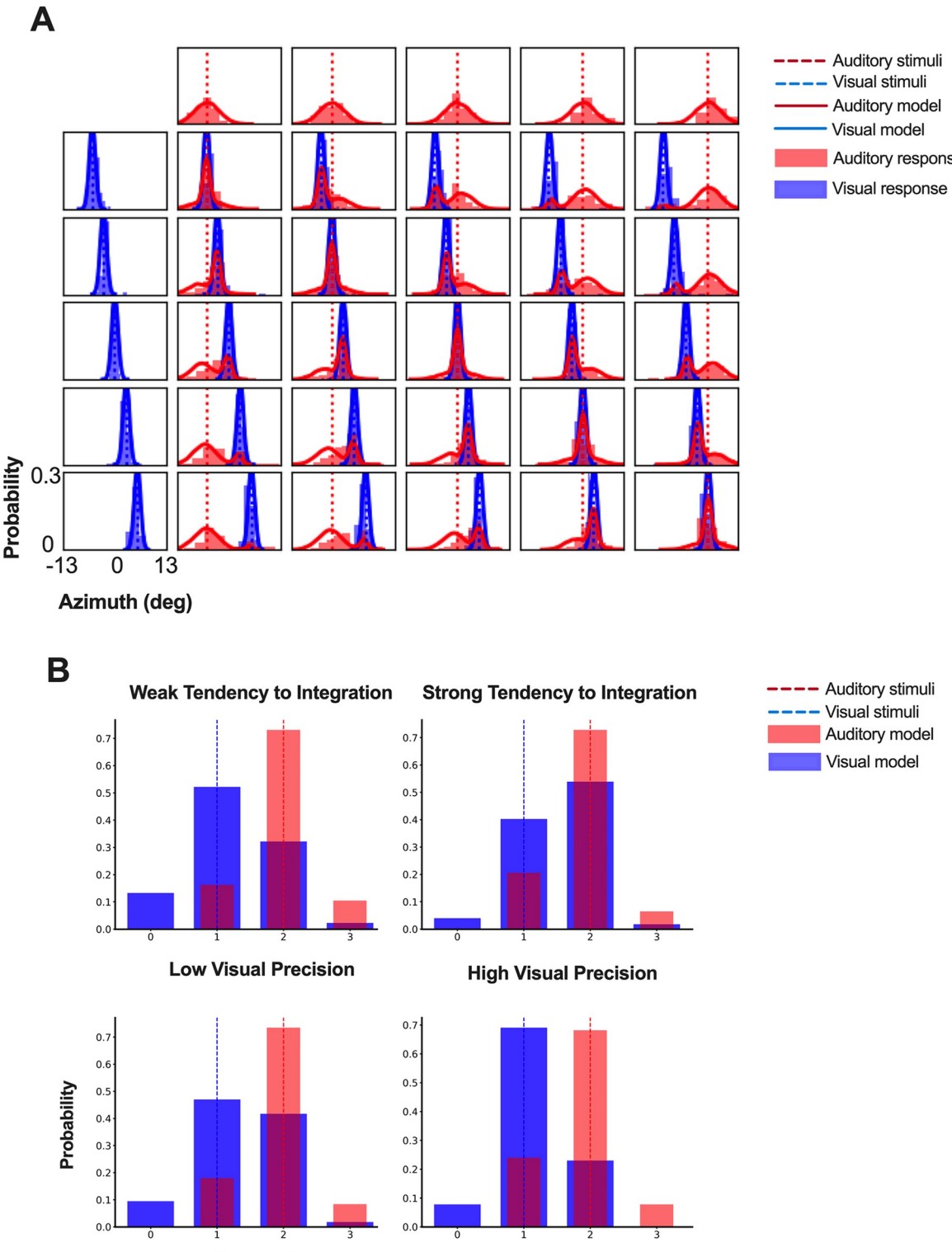

**Fig 4. Examples of BCI toolbox outputs.** (A) The model fitting results with continuous data from a spatial localization task. Each plot corresponds to one of the stimulus conditions, with the first row plots representing unisensory auditory conditions (stimulus position varying from left-most to right-most positions along azimuth from left to right), and first column representing unisensory visual conditions, and all other plots corresponding to bisensory conditions. Positions of the auditory and visual stimuli are denoted using broken red and blue vertical lines, respectively. The red and blue histograms represent the auditory and visual response distributions of a

specific subject, respectively. The red and blue solid lines represent the model fits produced by the toolbox. (B) The simulation results for one visual stimulus accompanied by two auditory stimuli. We used the fixed parameters (Weak tendency: $p_{common}$ = 0.2; Strong tendency: $p_{common}$ = 0.8; $\sigma_1$ = 1; $\sigma_2$ = 0.5; $\sigma_P$ = 1.5; $\mu_P$ = 1.5) to simulate how prior integration tendency influences multisensory numerosity perception. We also used the fixed parameters ($p_{common}$ = 0.5; Low visual precision: $\sigma_1$ = 1; high visual precision: $\sigma_1$ = 0.5; $\sigma_2$ = 0.5; $\sigma_P$ = 1.5; $\mu_P$ = 1.5) to simulate how unisensory precision influences multisensory numerosity perception.

estimated by the model with an error margin of 5% or less. The current work provides compelling evidence of the scientific validity and reproducibility of the BCI model, offering a reliable data processing option for future cognitive neuroscience research. We have made the BCI toolbox open source, and encourage researchers to extend and modify the BCI model based on their specific research needs.

## Potential limitations of BCI Toolbox

A notable constraint of the toolbox is the fixed number of variables in each model. Therefore, users might face challenges in scenarios where flexible configurations or customizations of variables are required, potentially hindering the adaptability of the tool for diverse research applications. Additionally, although current methods like computing log likelihood or sum of squared errors are used in the BCI Toolbox to measure model-behavioral data discrepancies, improvements are needed. The loss function in the brain is shaped by evolution or experience to minimize specific costs, which varies among individuals and over time [4]. Therefore, we will continue to explore additional possible methods of quantifying the error which may yield better fits. Updates to the BCI Toolbox and its documentation will be provided in due course to reflect these advancements [25].

In summary, the BCI Toolbox integrates the resources of cognitive neuroscience research that BCI models can interpret in the past decade. It applies the latest algorithms and parameter optimization methods, providing a convenient, reliable, and diverse data processing tool for potential studies. By utilizing top-notch datasets and cutting-edge models, the present work greatly enriches the computing community of cognitive neuroscience. We encourage fellow community members to contribute to its improvement by suggesting improvements, reporting bugs, and offering bug fixes, new ideas, and innovative modifications.

## Supporting information

**S1 Text. Supplemental Results.**
(DOCX)

**S1 Fig. The binding tendencies (Pcommon) from the pre-test and the post-test localization tasks.** The half-violin plot shows the distribution of the binding tendencies estimated through the BCI Toolbox. The purple and blue dots represent the individual subject Pcommon values for pre-test and post-test, respectively. The dotted lines link the pre- and post-test data, and the solid line links the mean values. Wilcoxon signed-rank test shows significantly different binding tendencies for the pre- and post-tests. *p = .005.
(TIFF)

**S1 Table. The optimized parameter values ± standard error estimated from behavioral data using the BCI Toolbox.** Wilcoxon signed-rank test shows significantly different binding tendencies (Pcommon) for the pre- and post-tests. *p = .005.
(DOCX)

## Acknowledgments

We thank Aijun Wang for helpful discussions, Saul I. Quintero and Kimia Kamal for testing the toolbox and their suggestions, and Yue Lin for designing the logo for the toolbox.

## Author Contributions

**Conceptualization:** Ulrik Beierholm, Ladan Shams.

**Formal analysis:** Haocheng Zhu, Ulrik Beierholm, Ladan Shams.

**Funding acquisition:** Ulrik Beierholm, Ladan Shams.

**Investigation:** Haocheng Zhu, Ulrik Beierholm, Ladan Shams.

**Methodology:** Haocheng Zhu, Ulrik Beierholm, Ladan Shams.

**Project administration:** Ulrik Beierholm, Ladan Shams.

**Resources:** Ulrik Beierholm, Ladan Shams.

**Software:** Haocheng Zhu, Ulrik Beierholm, Ladan Shams.

**Supervision:** Ulrik Beierholm, Ladan Shams.

**Validation:** Haocheng Zhu.

**Visualization:** Haocheng Zhu.

**Writing – original draft:** Haocheng Zhu, Ulrik Beierholm, Ladan Shams.

**Writing – review & editing:** Haocheng Zhu, Ulrik Beierholm, Ladan Shams.

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
