## [Decision Letter · Decision Letter 0]

18 Apr 2024

Dear Prof. Shams,

Thank you very much for submitting your manuscript "BCI Toolbox: An Open-Source Python Package for the Bayesian Causal Inference Model" for consideration at PLOS Computational Biology. As with all papers reviewed by the journal, your manuscript was reviewed by members of the editorial board and by several independent reviewers. The reviewers appreciated the attention to an important topic. Based on the reviews, we are likely to accept this manuscript for publication, providing that you modify the manuscript according to the review recommendations. We take the opportunity to  thank the authors for their patience.

The three reviewers offered constructive comments but all felt that the work was worthwhile.

My decision is for Minor Revisions to be undertaken. In this case, this means that you need to consider carefully the suggestions of the Reviewers, and to address their points in a detailed Response to Reviewers.

I would particularly encourage the authors to consider implementing changes to the software and especially its help and how-tos that will reduce the chance of erroneous usage, especially with regards to missing data and parameter recovery (Reviewer 1).

However, you do not need to change the software itself in response to every suggestion to do so, but to clearly argue in the Response to Reviewers which changes may be out of scope of your stated goals to implement.

Sincerely,

Michael Moutoussis

Guest Editor

PLOS Computational Biology

Thomas Serre

Section Editor

PLOS Computational Biology

We thank the authors for their patience regarding "BCI Toolbox: An open-source Python package for the Bayesian causal-inference model".

As you can see, our three reviewers offered constructive comments but all felt that the work was worthwhile.

My decision is for Minor Revisions to be undertaken. In this case, this means that you need to consider carefully the suggestions of the Reviewers, and to address their points in a detailed Response to Reviewers.

I would particularly encourage the authors to consider implementing changes to the software and especially its help and how-tos that will reduce the chance of erroneous usage, especially with regards to missing data and parameter recovery (Reviewer 1).

However, you do not need to change the software itself in response to every suggestion to do so, but to clearly argue in the Response to Reviewers which changes may be out of scope of your stated goals to implement.

Reviewer's Responses to Questions

**Comments to the Authors:**

Reviewer #1: Zhu and colleagues report on the development of an open-source toolbox for Bayesian Causal Inference (BCI), including a graphical user interface.

I am sympathetic to the fact that BCI has been put forward as a general unifying framework in neuroscience, and thus facilitating its implementation and scrutiny is a valuable step forward. I find the current toolbox of too narrow scope to justify publication in PCB, but I believe the authors can expand on it.

Shams & Beierholm, 2022, argue for BCI as a general neuroscience framework, and detail all the different tasks that depend on BCI. Figure 2 of this paper shows a number of different graphical models. I think for this toolbox to be truly useful, it would have to allow users to implement different graphical models. Ideally the users could draw their own graphical model. If this is not feasible, at least I think the authors could implement the graphical models shown in Figure 2 of Shams & Beierholm, 2022.

The BCI literature is very strong at the moment, with some beautiful cognitive neuroscience work (EEG/fMRI). Much of this work (e.g., Uta Noppeney and colleagues) employs representational dissimilarity matrices (RDMs) as derived from BCI to regress neural function (whether EEG or fMRI; eg., Rohe et al., 2019, Figure 2). I think it would be valuable to allow users to output RDMs that could then be used for further neural analysis.

Facilitating the use of BCI to users who are not programmers is a laudable goal. However, I worry that user-friendly tools as this one could be mis-used. Thus, I think the authors need to implement some quality control metrics/pipelines, as well as give potential future users (who will be reading this manuscript) some more guidance. For instance:

- What should be done with ‘missing data’?

- Could the authors provide a simulation demonstrating how parameter recovery changes with number of trial per condition, number of conditions, noise in the data, etc.

- I think the authors should also implement the ability to request full posteriors for individual fits. Acerbi has recently implemented a PyVBMC, which may reduce the computational burden in estimating full posteriors for each subject.

Future readers may also want an estimate of run-times as a function of trial numbers, subject numbers, etc.

Could the authors provide model recovery estimates when wrong assumptions are made? What happens if a user assumes model averaging when in fact subjects are performing model selection or probability matching?

I agree with the authors that facilitating the examination of BCI (also for scientists without programming skills) is valuable. I would expect many of those potential users being individuals with clinical expertise. Given the rise of computational psychiatry as well, I can see this manuscript and toolbox being used in clinical psychiatry, as such, I think it would be interesting to include in the introduction mention to recent papers arguing for computational psychiatry in general (Huys et al., 2016; Lawson et al., 2017; Van de Cruys et al., 2014; Karvelis et al., 2018) as well as a focus on casual inference (Noel et al., 2022, Noel & Angelaki, 2023)

Reviewer #2: Review of "BCI Toolbox: An open-source Python package for the Bayesian causal-inference model"

Reviewer: Michael Landy

This is a short report on a software package to allow people to explore and fit the standard causal-inference model in a plug-and-play fashion. This is a nice contribution to the field.

Comments (page/para/line; note that page numbers would have been helpful, and I'm treating the title page as page 1):

* The grammar could use the careful eye of a native speaker throughout

* I was unaware that PLoS Computational Biology accepts articles on methods or toolboxes, but I'm glad that they do.

* The current implementation is intended as a first pass with extension suggestions from the community welcome. The specific model implemented allows for two cues with values either at multiple equally-spaced discrete locations or continuous, a single, supra-modal Gaussian prior (I'm not sure what prior is allowed in the discrete case) and assumes a report for a single modality's location/number. As the authors are aware, my lab (uncited ;^) has done several studies involving fitting and model-comparison of this model, and all of that work involves elaborations that are not possible with the current toolbox. These extensions include: modality-specific priors (not exactly consistent with the original Bayesian logic) and non-Gaussian (e.g., bimodal) priors, affine miscalibration of one modality relative to the other (more general than Eqs. 1-2), models of the recalibration process for p_common and for cue calibration, allowing for common vs. separate-cause responses as well as localization responses, and multiple models of the causal-inference response.

4/1/5: "sampling the likelihood": It's important to be careful and precise about terminology in describing these models, as well as clearly specifying whether you are, at any given point in your text, describing things from the observer's perspective for a single trial vs. describing things from the experimenter's perspective for evaluating data from multiple trials. The generative model has something usually called the measurement distribution. This is from the experimenter's perspective of the model: the measurement distribution is p(x|s) where s is fixed and x is sampled. From the observer's perspective, they know x but don't know s, and p(x|s) seen this way is l(s|x), i.e., the likelihood of various stimulus locations given the noisy measurement. You can sample a measurement distribution, You can't sample a likelihood per se (it's not even necessarily a probability distibution, i.e., needn't integrate to one, although it does for your case of a fixed measurement SD independent of value; implementing Weber's Law for intensive variables will break this). So, please be careful with terminology for readers who aren't already familiar with the model.

Eqs. 1-2: In addition to being less general than our affine model of miscalibration, this is also too general, because one can only estimate gamma_A-gamma_V from data, not the individual gamma values.

5/near bottom: "minimize the error in the inference of causal structure": This makes no sense. The inference of causal structure is a binary inference (common vs. separate causes). You can be optimal in that (select the model whose posterior probability is greater than 0.5), where "optimal" is based on a cost function in which you want your probability of being correct in the causal inference to be maximized. But, that optimality criterion is irrelevant to the optimality criterion (i.e., cost function) you use for the localization judgment. Model selection says that you select the model first and then stop considering the model you've rejected. The resulting localization judgment is NOT based on an optimization of the localization judgment based on a cost function. Both model selection and probability matching are typically described as suboptimal heuristics.

6/2: "according to its inferred probability" is unclear. Easier to understand if you state overtly that you calculate the posterior probability of a common cause, then flip a weighted coin to decide randomly whether to report the separate- vs. common-cause location estimate.

10/2/5: "discrete data model": does this model have guiding parameters, or does it merely assume that, as for numerosity, the domain of judgments is the non-negative integers. Say so explicitly.

Figure 3: You don't say what the solid lines are. I assume they are regression lines, but who cares about regression for this. The lines should be identity lines and the reported values should be MSE, not R^2

12/1/10-12: You can't have only fit the pre-adaptation data if you replicated the increase in p_common (since that requires separately fitting the post-adaptation p_common). You might want to look at Hong et al. (YEAR) from my lab for a more elaborate take on modeling the effect of adaptation on p_common.

12/2: CHECK REF FOR ITS MODEL. This simulation is based on a model that takes two noisy discrete counts (of flashes and of beeps) and then does BCI to infer a number of each as if these are merely counts on a discrete scale with nothing else going on. My lab has also worked on this (only a VSS abstract from Steph Badde, sigh, and the subsequent modeling I'm done never got finished nor published, so of course you don't know about it). But, my point is that the model you are simulating doesn't take into account temporal discrepancy. If you hear a single beep and see a single flash, you won't infer common-cause based on the equality of the number of events if there is a 10 second discrepancy in arrival time!

16/2/3: "The toolbox also offers a model comparison option...": What model-comparison statistics do you implement?

Reviewer #3: This paper presents a toolbox for performing causal inference (CI), a model widely used to explain how our brains combine information.

While the mathematical challenges of performing inference in often complicated CI models make it less accessible, a toolbox like the one proposed in this paper can greatly improve it and increase its appeal to the community. The paper also provides the ability to use BCI for both simulation and parameter fitting, which can help any researcher explore the potential of a BCI model for their data, experiment, and scientific question. They contribute to the ease of use of BCI by providing a user-friendly GUI that packages established parameter learning modules, such as VBMC.

While the paper has helped to package different components of using BCI models under a user-friendly GUI, there are some aspects of the paper that can be better discussed or improved:

1) The paper primarily uses one-dimensional stimuli to test and validate the model. While I understand the complications of extending the work to multi-dimensional stimuli, it would be great to (1) have some simple special cases of multi-dimensional stimuli and their parameter recovery included, (2) if there are multi-dimensional experiments, can a certain set of assumptions still allow us to use the package. A detailed discussion of this issue can help researchers using the package and improve the potential of using it for complicated experiments.

2) The toolbox allows users to choose between different decision strategies, but I would hope to see an option that allows users to be informed about the best decision strategy based on their experimental data. Reports can be presented for all types of decision strategies, but a model comparison approach can be taken to inform users of the decision strategy that best explains their data.

3) In addition, allowing users to define specific loss functions or decision strategies would be very beneficial. For example, instead of using just one sample to do probability matching, the model could use N samples. I can see this being done by adding a way to write a custom Python snipper that takes in a posterior and outputs an answer. If this is not feasible, please mention in the discussion why this is hard and how it could be addressed in future work.

4) I had trouble getting the fitting to work. I simulated data and then imported it, but the run gave me a blank screen. It would be great to have a single demo button that sets parameters in the simulated data that can be exported and then fit. Maybe the toolbox al

---

## [Editor Report · Decision Letter 1]

5 Jun 2024

Dear Prof. Shams,

We are pleased to inform you that your manuscript 'BCI Toolbox: An Open-Source Python Package for the Bayesian Causal Inference Model' has been provisionally accepted for publication in PLOS Computational Biology.

Best regards,

Michael Moutoussis

Guest Editor

PLOS Computational Biology

Thomas Serre

Section Editor

PLOS Computational Biology

In my opinion, the authors have responded well to the reviewers' comments and conerns, changing the manuscript in line with suggestions that were in-scope. My decision as guest editor is to accept this manuscript for publication in PLOS-CB.

---

## [Editor Report · Acceptance letter]

28 Jun 2024

PCOMPBIOL-D-24-00004R1 

BCI Toolbox: An Open-Source Python Package for the Bayesian Causal Inference Model

Dear Dr Shams,

I am pleased to inform you that your manuscript has been formally accepted for publication in PLOS Computational Biology. Your manuscript is now with our production department and you will be notified of the publication date in due course.

With kind regards,

Olena Szabo
